# Efficacy of Five Disinfectant Products Commonly Used in Pig Herds against a Panel of Bacteria Sensitive and Resistant to Selected Antimicrobials

**DOI:** 10.3390/ani12202780

**Published:** 2022-10-15

**Authors:** Clara Montagnin, Shaun Cawthraw, Isaac Ring, Fabio Ostanello, Richard P. Smith, Rob Davies, Francesca Martelli

**Affiliations:** 1Istituto Zooprofilattico Sperimentale della Lombardia e dell’Emilia-Romagna, Via Bianchi 9, 25124 Brescia, Italy; 2Animal and Plant Health Agency, Woodham Lane, New Haw, Addlestone KT15 3NB, UK; 3Department of Veterinary Medical Sciences, University of Bologna, Via Tolara di Sopra, 50, 40064 Ozzano dell’Emilia, Italy

**Keywords:** disinfectant, biocide, antimicrobial resistance, AMR, MDR, minimum inhibitory concentration, MIC, minimum bactericidal concentration, MBC, bacteria

## Abstract

**Simple Summary:**

Antimicrobial resistance (AMR) is a growing concern worldwide in both human and veterinary medicine. Biosecurity and the ability to limit bacterial contamination on farms are crucial in the fight against disease, including AMR bacteria. This study aimed to investigate whether the recommended concentrations of five commercial disinfectants (A to E) were sufficient to inhibit growth and inactivate a panel of bacterial strains, including some that carry multidrug resistance to selected antimicrobials. The effectiveness of each disinfectant was expressed as the minimum inhibitory concentration (MIC) and the minimum bactericidal concentration (MBC). The results indicate that the type of disinfectant and its concentration influence the inhibitory and bactericidal efficacy. The glutaraldehyde/quaternary ammonium-compound-based (disinfectant D) and chlorocresol-based products (disinfectant B) were the most effective in this study. Varying results were observed for the other compounds, depending on the bacterial species tested. The iodine-based (disinfectant C) and potassium-peroxymonosulfate-based (disinfectant A) products were less able to inhibit or inactivate the bacteria. The disinfectant products were less effective against the strains of *E. coli* with different phenotypic profiles of antimicrobial resistance compared to sensitive strains.

**Abstract:**

The growing threat of antimicrobial resistance worldwide has led to an increasing concern in the human, veterinary, and environmental fields, highlighting the need for strategies to effectively control bacterial contamination. Correct biosecurity practices, including the appropriate use of disinfectants, play a crucial role in controlling bacterial contamination. This study aimed to verify whether the recommended concentrations defined according to the Defra General Orders concentration (GO, published by the UK Department for Environment, Food and Rural Affairs’ disinfectant-approval scheme) of five commercial disinfectant preparations (peroxygen-based, phenol-based, two halogen-releasing agents, and glutaraldehyde/quaternary ammonium compound-based; disinfectants A to E, respectively) were sufficient to inhibit growth and inactivate selected bacterial strains, including some that carry known phenotypic patterns of multidrug resistance. The effectiveness of each disinfectant was expressed as the minimum inhibitory concentration (MIC) and minimum bactericidal concentration (MBC) values, determined by the broth-microdilution method. The results indicate that the type of disinfectant and its concentration influence the inhibitory and bactericidal efficacy. The glutaraldehyde/quaternary ammonium compound-based (disinfectant D) and chlorocresol-based products (disinfectant B) were the most effective, and the GO concentration was bactericidal in all the strains tested. The efficacy of the other compounds varied, depending on the bacterial species tested. The GO concentrations were at least able to inhibit the bacterial growth in all the products and bacterial strains tested. A greater tolerance to the compounds was observed in the strains of *E. coli* with multidrug-resistance profiles compared to the strains that were sensitive to the same antimicrobials.

## 1. Introduction

Antimicrobial resistance (AMR) is a natural phenomenon that has been amplified by the selective pressure exerted by the extensive use of antimicrobial treatments in humans and animals. Over the last decades, AMR has risen rapidly and is now considered one of the top 10 global health threats facing humanity [1]. Moreover, while the production of new therapeutic compounds to replace those that have lost their efficacy is proceeding slowly, the speed of antimicrobial-resistance development and the spread of multi- and pan-resistant bacteria seem to be accelerating [2].

As a consequence, the strengthening of biosecurity practices and the appropriate use of biocides have gained pivotal roles in infection control and the preservation of animal and human health [3]. A positive correlation between enhanced cleaning and disinfection (C&D) procedures and reductions in antimicrobial usage has already been observed in previous studies [4].

“Biocide” is a generic term for a chemical product capable of inactivating microorganisms [5]. Disinfectants are biocidal compounds widely used in primary production and they are crucial in the internal biosecurity procedures applied in intensive farming. The effective use of biocides plays a crucial role in disease control through C&D procedures, especially when applied as part of all-in/all-out management.

Evidence shows that exposure to sub-inhibitory concentrations of disinfectants can enable some bacteria to develop increased tolerance to disinfectants and even cross-resistance to antimicrobials [5,6]. However, it is believed that biocides are unlikely to induce such cross-resistance if used correctly, as their mechanisms of action are mainly non-specific and multifactorial. Disinfectants often consist of chemicals and mixtures of active components that are toxic towards different microbial targets [5,7]; therefore, the presence of resistance genes or mutations within a single gene is generally not sufficient to confer resistance in most bacteria, except in the case of quaternary ammonium compounds [8]. Furthermore, their recommended concentrations are generally well above the minimum inhibitory concentrations (MICs) [9].

Nonetheless, bacteria can develop tolerance towards disinfectants when the concentrations are below those required for a bactericidal effect [10]. Sub-optimal concentrations can result from excessive dilution, the presence of residual wash-water/detergents, organic matter or biofilms [11,12].

Several in vitro experimental tests have assessed the correlation between antimicrobial-resistances and disinfectant tolerances [13,14]. Others have shown that exposure to sub-lethal concentrations of disinfectants could select mutants with reduced sensitivity to antimicrobials due to co-selection [15].

In the UK, the Department for Environment, Food and Rural Affairs (Defra) maintains a list of approved disinfectants suitable for use when an outbreak of notifiable disease occurs [16]. Disinfectants can be approved for use at specified concentrations, according to five different specified protocols: TB (Bovine Tuberculosis) Orders [17], FMD (Foot-and-Mouth disease) Orders [18], SVD (swine vesicular disease) Orders [19], Diseases of poultry Orders [20,21], and General Orders (GO) [22]. GO is the concentration recommended for use against pathogens that do not have their own specific approval.

The GO approval test is performed by the Animal and Plant Health Agency (APHA, Addlestone, UK) using National Collection of Type Cultures *Salmonella* Enteritidis strain 13665 (NCTC 13665) as the challenge organism. The in vitro test is considered passed when the *Salmonella* concentration is reduced by at least five logs after exposure to the disinfectant for 30 min at 4 °C in the presence of yeast as the interfering substance [23].

The aim of this work was to verify whether the recommended GO concentrations for five different disinfectants (as published in March 2020) were sufficient to inhibit growth or inactivate a range of bacterial strains, some of which have known phenotypic patterns of antimicrobial-resistance. The broth-microdilution method used in this study differs from the GO test performed by APHA, mainly due to the absence of a soiling agent (yeast) simulating the presence of organic matter. The lack of an interfering substance represents a best-case scenario for disinfectants and is therefore likely to have positively influenced their overall performance.

## 2. Materials and Methods

### 2.1. Bacterial Strains

The minimum inhibitory concentration (MIC) and the minimum bactericidal concentration (MBC) of five disinfectant products were determined for 15 bacterial strains.

Ten genetically distinct strains of *Escherichia coli* were included in the panel, with a range of AMR profiles. The antimicrobial-resistance profiles of these AMR *E. coli* strains, as determined by MIC according to the specifications of Commission Implementing Decision 2013/652/EU, are shown in Table 1. For subsequent analysis, multidrug resistance (MDR) was defined as resistance to three or more antimicrobial classes [24]. Of these strains of *E. coli*, five were MDR: one was an extended spectrum β-lactamase/AmpC β-lactamases (ESBL/AMPC) producer (4536), one was an ESBL producer (4512), and one was an AMPC producer (4534).

These isolates were obtained from different UK pig farms enrolled in previous studies and stored in frozen beads at the Bacteriology department at APHA.

The other five bacteria screened, all from the UK National Collection of Type Cultures (NCTC), were: *Pseudomonas aeruginosa* (NCTC 13359), *Staphylococcus aureus* (NCTC 10788), *Proteus vulgaris* (NCTC 4175), *Enterococcus hirae* (NCTC 13383), and *Salmonella* Enteritidis (NCTC 13665).

### 2.2. Disinfectants

Five disinfectants (disinfectants A to E), purchased from commercial suppliers, were tested against the fifteen bacterial strains. All the disinfectants tested are typically intended for veterinary applications, including disinfection of buildings and transport vehicles; thus, should to be effective against the bacterial strains included in the study.

The active ingredients of the disinfectants and their recommended concentrations (GO) are shown in Table 2.

Disinfectant A is a broad-spectrum granular disinfectant, which is activated when added to water. It consists of a stabilized blend of a peroxygen compound (potassium peroxymonosulfate, KMPS), a surfactant (sodium dodecyl benzene sulphate), organic acids (malic and sulphamidic acids), an inorganic buffer (sodium hexameta phosphate), and sodium chloride. The main active component is KMPS, which oxidizes proteins of the bacterial cells, destroying their physical structure. Its antibacterial activity is synergized/enhanced by sodium chloride, which reacts with the KMPS and releases another powerful biocide, hypochlorous acid.

Disinfectant B is a chlorocresol-based disinfectant. Cresol is active both on bacterial membranes and on bacterial cytoplasm: at low concentrations, it causes loss of membrane integrity, while at higher concentrations, it has a coagulative effect on cytoplasmatic constituents [25,26].

Two different halogen-releasing disinfectants were also tested: disinfectants C and E. Disinfectant C contains a mixture of iodophors, sulfuric acid, and phosphoric acid. The bactericidal activity is mediated by the free molecular iodine (I_2_), stabilized and continuously generated by iodophors [5,26]. Iodine kills cells by inhibiting protein function and reacting with nucleotides and fatty acids [5,26,27].

Disinfectant E is an effervescent water-soluble chlorine tablet containing troclosene sodium (NaDCC) as the active compound. Once dissolved in water, it releases the active form of chlorine. The exact mechanism of action of this type of disinfectant has not been fully clarified, but the inhibition of some enzymes responsible for key reactions for cell life, such as the replication cycle or protein synthesis, has been hypothesized [28].

Disinfectant D consists of a mixture of four different groups of active compounds: glutaraldehyde, a combination of quaternary ammonia (QAC), alcohol (isopropanol), and pine oil, along with buffering agents and stabilizers. QACs are cationic surfactants that cause generalized membrane damage [5,26], while glutaraldehyde is an alkylating agent and causes inter- and intra-protein cross-linkage.

Each disinfectant was diluted in Mueller Hinton broth (MHB) up to 4 times the GO concentration.

Disinfectant E was insoluble in MHB alone and the precipitation of the disinfectant solid phase in the MIC microtiter plate made the bacterial growth difficult to assess. Therefore, this compound was diluted in an equal volume of MHB and water and left at 4 °C for an hour before the test begun.

### 2.3. Determination of the Minimum Inhibitory Concentration (MIC)

Both the bacteriostatic (MIC) and bactericidal (MBC) ability were evaluated. MIC and MBC values were expressed as the geometric mean of dilution rates of 6 and 12 replicates, respectively.

The MIC values of each disinfectant formulation against the selected isolates were determined by the broth-microdilution method [29]. Colonies from overnight growths of the bacterial strains were picked with a sterile loop and suspended in sterile demineralized water (Thermo Scientific, Loughborough, UK). The bacterial suspension was vortexed and adjusted using a nephelometer (Thermo Scientific, Loughborough, UK) for automatic turbidity measurement until an optical density (OD) of 0.5 nephelometric turbidity unit (NTU) on the McFarland scale was achieved.

Twenty μL of bacterial suspension were diluted in 11 mL of MHB with TES (Thermo Scientific, Loughborough, UK), giving an initial concentration of approximately 10^5^ CFU/mL. To ensure the accuracy of the bacterial concentration, 10^2^ and 10^3^ dilutions of the suspensions were made and plated onto 5% Sheep Blood Agar. After 24 h incubation, the colonies were counted, and the concentrations calculated.

Initially, each disinfectant was diluted in MHB to a concentration of 4 times the GO recommended concentration. If the MIC and MBC could not be determined, lower or higher starting concentrations were tested.

One hundred μL of each 4× disinfectant/MHB dilution were then pipetted in triplicate into the first row of a 96-well, round-bottomed microtiter plate (Thermo Scientific Nunc, Loughborough, UK). Serial, two-fold dilutions in MHB were made in rows of 2−8 using 50 μL from the first row added to the second row, etc. (discarding 50 μL from the final dilution). To each well, 50 μL of bacterial inoculum (approximately 10^5^ CFU/mL) were then added, to give a total volume of 100 μL (further diluting the disinfectants 1:1). The concentrations were adjusted where necessary in subsequent testing for some bacteria/disinfectants if MIC and MBC values were outside the initial dilution range.

For disinfectant D, since bacterial growth was inhibited at very high dilutions, different starting concentrations were used, chosen on the basis of the results obtained during the study. The initial concentrations of the disinfectants are shown in Table 2.

As a positive control, 50 μL of each bacterial suspension were added to 50 μL of MHB without disinfectant. The plates were then incubated for 18−24 h at 37 °C. After incubation, bacterial growth was assessed visually using a Sensititre Vizion instrument (Thermo Scientific, Loughborough, UK) and the lowest concentration without an observable ‘button’ of bacterial growth was established as the MIC value.

For all the disinfectant/bacterial strain combinations, three replicates were performed on two separate occasions (for a total of 6 MIC values).

### 2.4. Determination of the Minimum Bactericidal Concentration (MBC)

In order to determine MBC values, duplicate 10-microliter aliquots from each well of the microtiter plate were plated onto blood agar plates [29]. Once dried, the plates were incubated for 18−24 h and read. The lowest concentration with no visible bacterial growth was recorded as the MBC value (approximately 3.5 log reduction in the number of CFUs).

For each disinfectant/bacterial strain combination, six replicates were performed on two separate occasions (for a total of 12 MBC values).

### 2.5. Statistical Analysis

The Kolmogorov-Smirnov test (K-S) for goodness of fit was used to verify normality of the MIC and MBC data distribution. After confirming an abnormal distribution, the Mann–Whitney U-test was used to compare MIC and MBC values between Gram+ and Gram− bacteria and between MDR and non-MDR *E. coli* strains.

Since all MIC values were lower than or equal to the GO and the bactericidal effect is of particular relevance for disinfection, the microorganisms were divided into four categories on the basis of MBC values: “tolerant (T)”, when the MBC was higher than the GO (MBC > GO); “sensitive (S)”, when the MBC was lower than or equal to the GO (MBC ≤ GO); “particularly sensitive (PS)”, when the MBC was lower or equal than 2 × GO (MBC ≤ 2 × GO); and “extremely sensitive (ES)”, when the MBC was lower than or equal to 3 × GO (MBC ≤ 3 × GO). To evaluate the absolute difference between MIC and MBC values, the MIC/MBC ratio was also calculated, using the dilution values.

Analyses were conducted using SPSS software, version 25 (IBM SPSS, Armonk, NY, USA). Values of *p* < 0.05 were considered statistically significant.

## 3. Results

In all the disinfectant/bacterial-strain combinations, the geometric mean of the MIC was always lower than the GO (recommended-use concentration), except for the disinfectant E, for which four of the non-MDR *E. coli* strains, two MDR *E. coli* strains, and *P. aeruginosa* had a MIC that was equal to the GO concentration. However, for disinfectant A, disinfectant C, and disinfectant E, the MBC determined in vitro was higher than the GO for some of the bacterial strains (Table 3, Appendix A, Figure 1, Figure 2 and Figure 3).

### 3.1. Minimum Inhibitory Concentration (MIC) and Minimum Bactericidal Concentration (MBC)

For all the disinfectants tested, the MIC values were lower than or equal to their recommended-use concentration (MIC ≤ GO), both in their absolute values and in their geometric means of six replicates. To obtain a bactericidal action, higher concentrations of disinfectant are required than those that determine the bacteriostatic effect. Not all the disinfectants evaluated were able to inactivate the tested bacteria at a concentration lower than or equal to that recommended for their routine use (GO), and higher concentrations were needed for a bactericidal effect (MBC > GO).

Overall, 9 of the 15 (60%) bacterial strains tested showed tolerance to the bactericidal action at the GO concentration of at least one disinfectant: *P. aeruginosa* and *S. aureus* demonstrated tolerance to one disinfectant each (disinfectant E and disinfectant C, respectively), and *P. vulgaris*, *S.* Enteritidis, and *E. coli* strain MSG17 C20 showed tolerance to both disinfectant A and C. All five MDR *E. coli* strains showed tolerance to the bactericidal action of disinfectants A and C. Four of the five MDR strains showed tolerance to the bactericidal action of disinfectant E. By contrast, all the non-MDR *E. coli* and *E. hirae* were inactivated by all the commercial products and at concentrations lower than or equal (as in the case of disinfectant E) to the GO.

Disinfectant B and disinfectant D showed both bacteriostatic and bactericidal action at the concentration defined by their GO for all the bacterial strains tested. Although the MIC and the MBC were always higher than the GO, differences were highlighted between the different bacterial species and between the different strains of *E. coli*.

Disinfectant B (GO: 1:50) was, after disinfectant D (*n* = 15), the product with the second most “extremely sensitive” number of strains (*n* = 10). Together with disinfectant A, it was the product with the greatest MIC/MBC ratio. The MIC and MBC values for *E. hirae* (1:1796 and 1:1131, respectively) and *S. aureus* (1:5702 and 1:673, respectively) were the lowest of all the bacteria. For *S. aureus*, the greatest variation between the MIC and the MBC was also recorded (MIC/MBC = 8.48) (Figure 1, Appendix A). *Proteus vulgaris* and *P. aeruginosa* (MIC 1:898 and 1:1425, respectively) were efficiently inhibited by disinfectant B, while their inactivation concentrations were at least five times higher (MBC 1:141 and 1:283). The MIC and MBC values obtained from exposure of non-MDR *E. coli* strains to disinfectant B were similar for all the strains (1:504 < MIC < 1:400; 1:400 < MBC < 1:283). By contrast, the MIC/MBC ratio varied from 2.12 to 6.73 for the MDR strains (Figure 1, Appendix A). The five MDR *E. coli* strains showed the greatest tolerance to the bactericidal action of disinfectant B: although they were defined as “sensitive” (to disinfectant), the geometric mean of the MBC values obtained for one of them (strain 4534) was equal to the GO (1:50).

Disinfectant D had a greater inhibiting or bactericidal effect from the other products, even at dilutions of over 4000 times higher than the GO-approved concentration (1:33). Among all the bacteria, the lowest MIC and MBC values were found to be those for *S. aureus* (MIC = MBC 1:135,168) and *P. aeruginosa* (MIC = MBC 1:67,584). None of the bacterial strains tested showed tolerance to the bactericidal action of disinfectant D, to the extent that the highest MBC was about 100 times lower than the GO (*S.* Enteritidis MBC 1:3364) and all the bacteria were defined as “extremely sensitive” (MBC ≤ 3 × GO). The MIC value was equal to the MBC value (MIC = MBC) for 8 of the 15 bacterial strains tested: *P. vulgaris* (1:33,792), *S. aureus* (1:135,168), *P. aeruginosa* (1:67,584), and all five of the non-MDR *E. coli* strains (strain 25922 1:33,792; strains 4526 and 4527 1:23,895; strain 4529 1:21,288; strain 4531 1:16,896). Only *S.* Enteritidis had a greater difference between its MIC and MBC values (1:22,627 and 1:3364 respectively; MIC/MBC = 6.73) (Figure 2, Appendix A).

*Staphylococcus aureus* and *P. aeruginosa* were also “extremely sensitive” to disinfectant A (MIC: *S. aureus* 1:6,400, *P. aeruginosa* 1:2263; MBC: *S. aureus* 1:6041, *P. aeruginosa* 1:1,695). The inhibitory and bactericidal action of disinfectant A at its recommended-use concentration at the time of the study (1:100) was also effective against *E. hirae* (MIC 1:200; MBC 1:159), although the effectiveness was not as high as that observed for disinfectant B. The MIC value of *P. vulgaris* (1:504) was one of the lowest recorded for this disinfectant; however, since the MBC value (1:94) was higher than the GO (1:100), it was classified as “tolerant”. Furthermore, for the five non-MDR *E. coli* strains, the MIC value was equal to the MBC value (MIC = MBC) and was, for all but one (strain 4526, MIC = MBC 1:283), equal to 1:200 (Figure 1, Appendix A). However, the 200-fold dilution (slightly higher for *E. coli* strain 4526, MIC 1:283) of disinfectant A was still able to inhibit all the non-MDR *E. coli* strains tested and kill susceptible strains. *Salmonella* Enteritidis (MBC 1:53) and the five MDR *E. coli* strains showed tolerance (MBC > GO) to the bactericidal effect of disinfectant A at its recommended dilution (strain 4512, MBC 1:26; strain 4534, MBC 1:25; strain 4536, MBC 1:40; strain 2229, MBC 1:26; strain MSG17 C20, MBC 1:53) and a greater difference between the MIC and the MBC, which was always at least 3.78 times greater than the MIC observed for the MDR *E. coli* strains (Figure 1, Appendix A). In general, together with disinfectant B, disinfectant A was the compound for which the difference between the MIC and MBC values was greatest but, unlike that of disinfectant A, the MBC of disinfectant B never exceeded the GO value (Figure 1, Appendix A).

The greatest prevalence of tolerance to bactericidal action by the bacterial strains was observed for disinfectant C (GO 1:49). While the MIC was consistently lower than the GO for all the bacterial strains examined (Figure 2, Appendix A), eight of the fifteen strains were “tolerant” (MBC > GO) of its bactericidal action, including *S. aureus* (MBC 1:41), which was “particularly sensitive” or “extremely sensitive” to all the other disinfectants. The other bacteria with MBC > GO were: *P. vulgaris* (MBC 1:35), *S.* Enteritidis (MBC 1:37), and the five MDR *E. coli* strains (strain 4512, MBC 1:37; strain 4534, MBC 1:28; strain 4536, MBC 1:29; strain 2229, MBC 1:28; strain MSG17 C20, MBC 1:29). All five non-MDR *E. coli* strains had MIC values lower than the GO. In addition, the MBC value was below the GO. *Enterococcus hirae* and *P. aeruginosa* both expressed MBC values of 1:156, which allowed them to be classified as “extremely sensitive”, together with the *E. coli* strain 25922 (MBC 1:196). It is interesting to observe how, for disinfectant C, the MIC values of *S. aureus* (1:69) and *P. vulgaris* (1:55) were very close to the GO (1:49). This was not observed for disinfectant A and disinfectant B, for which the effective inhibitory dilution was always at least double that expected by the GO. Disinfectant C was the product showing the least variation in the range of its MIC and MBC values towards the different bacterial species (1:55 ≤ MIC ≤ 1:196; 1:28 ≤ MBC ≤ 1:196).

For disinfectant E, the MIC value corresponded to the GO (1:360) for seven of the fifteen bacterial strains examined (including four of the five non-MDR *E. coli* strains, two of the MDR *E. coli* strains and *P. aeruginosa*), while for three bacteria, the dilution rate used did not even reach double that foreseen by the GO (MDR *E. coli* strain 4534, 1:509; MDR *E. coli* strain 2229, 1:404; *E. hirae*, 1:404). For five of the fifteen bacterial strains, the MBC exceeded the recommended-use concentration. Specifically, this occurred for *P. aeruginosa* (1:240) and for four MDR *E. coli* strains (strains 4512, 4536 and 2229, MBC 1:255; strain 4534, MBC 1:286). For four non-MDR *E. coli* strains (4526; 4527; 4529; 4531,) the MIC and MBC values coincided and were equal to the GO (MIC = MBC 1:360), while the MBC of strain 25922 was 1:454. *Enterococcus hirae*’s MBC was equal to the GO (1:360), and only *P. vulgaris* and *S. aureus* were defined as “extremely sensitive” (MBC 1:1814) and “particularly sensitive” (MBC 1:720), respectively, to disinfectant E. Finally, the lowest variability between MIC and MBC values was also observed for this disinfectant, and for *S. aureus*, the MBC was double that of the MIC.

Lastly, significantly greater sensitivity to both the bacteriostatic effect (MIC) of the Gram+ bacteria (*E. hirae* and *S. aureus*) compared to the Gram− bacteria was observed for three of the products tested: disinfectants A (*p* = 0.004), B (*p* < 0.001), and E (*p* = 0.034). This difference was also significant with regards to the bactericidal effect (MBC) and for the same commercial disinfectants: disinfectants A (*p* < 0.001), B (*p* < 0.001), and E (*p* = 0.011).

In summary, nine of the fifteen bacterial strains examined showed tolerance to bactericidal action at the GO concentration for at least one disinfectant, while *E. hirae* and all the non-MDR *E. coli* strains (25,922, 4526, 4527, 4529, and 4531) were inhibited by all the disinfectants. *Pseudomonas aeruginosa* and *S. aureus* expressed the highest sensitivity to the greatest number of disinfectants and expressed tolerance to a single device (disinfectants E and C, respectively).

### 3.2. Differences in MIC and MBC between Multidrug-Resistant and Non-Multidrug-Resistant Escherichia coli Strains

Any differences in MIC and MBC values were evaluated within the ten *E. coli* strains used belonging to the two different categories (five MDR strains and five non-MDR strains). Disinfectants A, B, C, and D inhibited all the strains at concentrations lower than the GO (MIC < GO), while disinfectant E inhibited four non-MDR strains (strains 4526, 4527, 4529, and 4531) and two MDR strains (strains 4512 and 4536) at concentrations equal to the GO (MIC = GO).

Only disinfectant B and disinfectant D had effective bactericidal action (MBC ≤ GO) against all the MDR *E. coli* strains.

Disinfectant B showed MIC and MBC values lower than or equal to the GO against all the strains; however, the MIC and MBC values of the MDR *E. coli* strains were significantly higher (*p* = 0.003 and *p* < 0.001, respectively) than the non-MDR *E. coli* strains. A significant difference (*p* < 0.001) in MBC values alone was also observed for disinfectants A, D, and E, against which the MDR *E. coli* strains were proven to be tolerant (MBC > GO). For disinfectant C, the MIC and MBC values were significantly different (*p* < 0.001) in the two groups of *E. coli* and higher for the MDR strains.

## 4. Discussion

The aim of this work was to verify in vitro the effectiveness of five different disinfectants against fifteen different bacterial strains and to determine whether their recommended concentration of use (GO) was sufficient to inhibit and kill a selection of bacterial species. In addition, for the *E. coli* strains, the possible association between the MDR characteristics and reduced sensitivity to disinfectants was evaluated.

Correct C&D procedures are essential components of any effective strategy for controlling microorganisms on livestock farms [30].

It is assumed that the efficacy of a disinfectant depends mainly on its mechanism of action, but it is critical to consider other factors, such as its concentration and the presence of organic matter on the treated surfaces.

One of the limitations of the available literature regarding disinfectants is the fact that many studies evaluate the effectiveness of disinfectants in the food industry or in hospital environments [31]. Furthermore, only a limited number of countries, including the UK, have developed and used official methods for the evaluation of disinfectants.

The MIC and MBC evaluation of disinfectants through broth dilution is easy to perform and standardize. However, it represents the best-case scenario for the disinfectant, because of the absence of an interfering substance. The performance of the disinfectant is therefore not inhibited and the concentration needed for its actual use is likely to be higher than the MIC and MBC.

Breakpoint values for biocide tolerance are not available [25,32]. For this reason, when the term “tolerant” is used in relation to biocides, it indicates a greater tolerance to, and a decrease in the susceptibility of a given microorganism to a particular concentration of disinfectant, to which it was normally susceptible [25,33].

In this study, none of the commercial disinfectants analyzed required lower dilutions than those defined by the GO to inhibit the growth of the selected bacterial strains examined.

The Gram+ bacteria included in this study showed, in most cases, greater sensitivity to disinfectants. *Enterococcus hirae* was inhibited and inactivated by all the products tested, even when strongly diluted. Similarly, *S. aureus*, despite being tolerant to the bactericidal action of disinfectant C, was “particularly sensitive” or “extremely sensitive” to all the other disinfectants examined. This was probably due to the nature of the membranes of these microorganisms. Gram+ bacteria generally have much less resistant cell membranes than Gram− bacteria, which instead offers a significant barrier, which limits the entry of active compounds and the consequent achievement of their target sites [5].

Another important point was that only disinfectant D and disinfectant B were shown to be bactericidal at their GO concentration, or lower, against all the bacterial strains tested. In particular, disinfectant D had both a bacteriostatic effect and bactericidal effect even at very high dilutions, well above the GO, so much so that the highest MBC value observed against *S.* Enteritidis was still about 100 times lower than the recommended concentration. Disinfectant B, on the other hand, required much higher concentrations to be bactericidal than those necessary to inactivate most of the bacteria, including *S. aureus*, *P. vulgaris*, *P. aeruginosa*, and the four MDR *E. coli* strains. Furthermore, very often, the MIC value coincided with that of MBC.

The results of this study therefore confirm the high efficacy against bacterial microorganisms of the quaternary ammonium compound/glutaraldehyde-based combination, but also of chlorocresol, already highlighted in other experimental work, even in the presence of organic matter [26,29].

In contrast to the two previous products, the results obtained indicate that the least effective in vitro disinfectants were those based on iodine (disinfectant C) or potassium peroxymonosulfate (disinfectant A). The *S. aureus* required an MBC higher than the GO for disinfectant C alone, while the remaining seven of the fifteen disinfectant-C-tolerant bacteria also showed tolerance to disinfectant A, including *S.* Enteritidis. It should be noted that the GO concentration of disinfectant A has changed to 1:80 since this study was conducted.

Payne et al. [34] found that potassium peroxymonosulfate (the active component of disinfectant A) had a high disinfectant efficacy against *Salmonella* under laboratory conditions. However, many studies have highlighted the poor efficacy of biocides with an oxidative mechanism and those based on iodine, both in vitro and in conditions that simulate those in the field [26,29]. Indeed, Martelli et al. [35] found residual contamination by *Salmonella* when, for the C&D procedures, an iodine-based product was used at a concentration higher than that recommended for use.

There may be many mechanisms of bacterial tolerance to disinfectants and it seems that *Salmonella* has developed a high tolerance to oxidative damage as a result of its ability to survive inside macrophages [36]. In line with previous research, high efficacy was observed in disinfectant D and disinfectant B, which were confirmed to be the most effective disinfectants against this bacterium [26,29,34]. The only observed tolerance of *P. aeruginosa* to inactivation was against disinfectant E, relating to four of the five MDR *E. coli* strains. This finding was quite unexpected, considering that the NaDCC (the active component of disinfectant E) is able to pass through the strongly hydrophobic membrane of *P. aeruginosa*, responsible for its high tolerance to many biocides, thanks to its low molecular weight [5,37]. Disinfectant E is a commercial disinfectant, sold in the form of effervescent tablets. It can be used both for the disinfection of surfaces and for the disinfection of drinking water thanks to the general effectiveness, ease of use, and stability of NaDCC [38].

*P. aeruginosa* and, especially, *E. coli*, are among the most commonly isolated bacteria amongst water contaminants, and they also have the ability to form biofilms, even in an aqueous environment, which offer protection against biocides and create the optimal conditions for their replication [39]. For this reason, the lack of effectiveness of disinfectant E against *P. aeruginosa* and most of the MDR *E. coli* strains (4 out of 5) is of concern, especially in the presence of biofilms.

The results obtained for the other bacteria confirm the disinfectant efficacy of disinfectant E. However, it is important to underline that for *E. hirae* and for the majority of non-MDR *E. coli* strains, the MIC and MBC values corresponded to the GO. The GO test does not claim to determine the bactericidal activity of a disinfectant; however, due to the absence of an interfering substance in this model, these results are of note, since the GO may be below the MIC and MBC threshold in field conditions.

Statistically significant differences in MIC and MBC values were found between the MDR and non-MDR *E. coli* strains. The strains with known MDR profiles exhibited reduced sensitivity to disinfectants, which resulted in increased MIC, MBC, or both, when compared to their antimicrobic-sensitive counterparts.

The possibility of cross-resistance between antimicrobials and disinfectants has already been highlighted previously [40], and it is known that the exchange of genes carrying tolerance to disinfectants and resistance to antimicrobials can be coupled (co-resistance) [15,41].

According to some research, an increase in the MIC of some antimicrobials could be due to the exposure of the bacteria to sub-inhibitory concentrations of commercial disinfectants [11,42], while other studies have found no significant correlation [43] and, in some cases, a negative association between antimicrobial resistance and reduced sensitivity to disinfectants [6,44].

The aim of the present study was not to evaluate a possible association between MDR and a reduced sensitivity to disinfectants. Nonetheless, the inclusion of MDR strains in the bacterial populations was useful to evaluate the possible presence of an association between disinfectant tolerance and resistance to antimicrobials, which will be further investigated in subsequent research.

The methods of definition of the MIC and MBC for the bacteria in suspension used in this study are relatively standardizable and produce repeatable and easily comparable results. However, the results obtained should be treated with caution as they refer to in vitro conditions. Field conditions involve several factors (e.g., the presence of organic matter, type of material of the treated surfaces, temperature, presence of biofilm, poor preliminary cleaning, and short time the disinfectant remains on the surfaces) that can influence the bacteriostatic and bactericidal efficacy of disinfectants. Furthermore, suspended bacteria are often more sensitive to the action of disinfectants than the bacteria present on dry surfaces [6]. It has often been shown that the presence of fecal material can affect the efficacy of biocides, even when used at recommended concentrations [5,45,46], such that incorrect or incomplete procedures for cleaning and removing organic matter can nullify the effectiveness of disinfectants. A more in-depth study would therefore be desirable, including the addition of organic soiling agents, such as bovine serum albumin (BSA), to the suspension in order to simulate the presence of organic matter and check for any variation in the concentration of effective disinfectant. Although the addition of BSA is not the preferred choice to simulate the contamination condition of farms, it would still be a fair compromise, given the ease of the method’s standardization and the difficulty of finding large quantities of sterile organic matter [47].

## 5. Conclusions

This study highlighted the existence of differences between commercial disinfectants intended for veterinary applications in the ability to inhibit and inactivate different bacterial populations. A lower sensitivity to disinfectants was also highlighted in the MDR *E. coli* strains than in the non-MDR strains. The most effective biocides were found to be disinfectant D and disinfectant B, while the iodine-based compound (disinfectant C) and the peroxymonosulfate-based compound (disinfectant A) proved to be the least effective. In conclusion, when carrying out C&D procedures on farms, it is crucial to choose the most active disinfectant for the microbial contaminants present and use it at the appropriate concentration to inactivate them.

## Figures and Tables

**Figure 1 animals-12-02780-f001:**
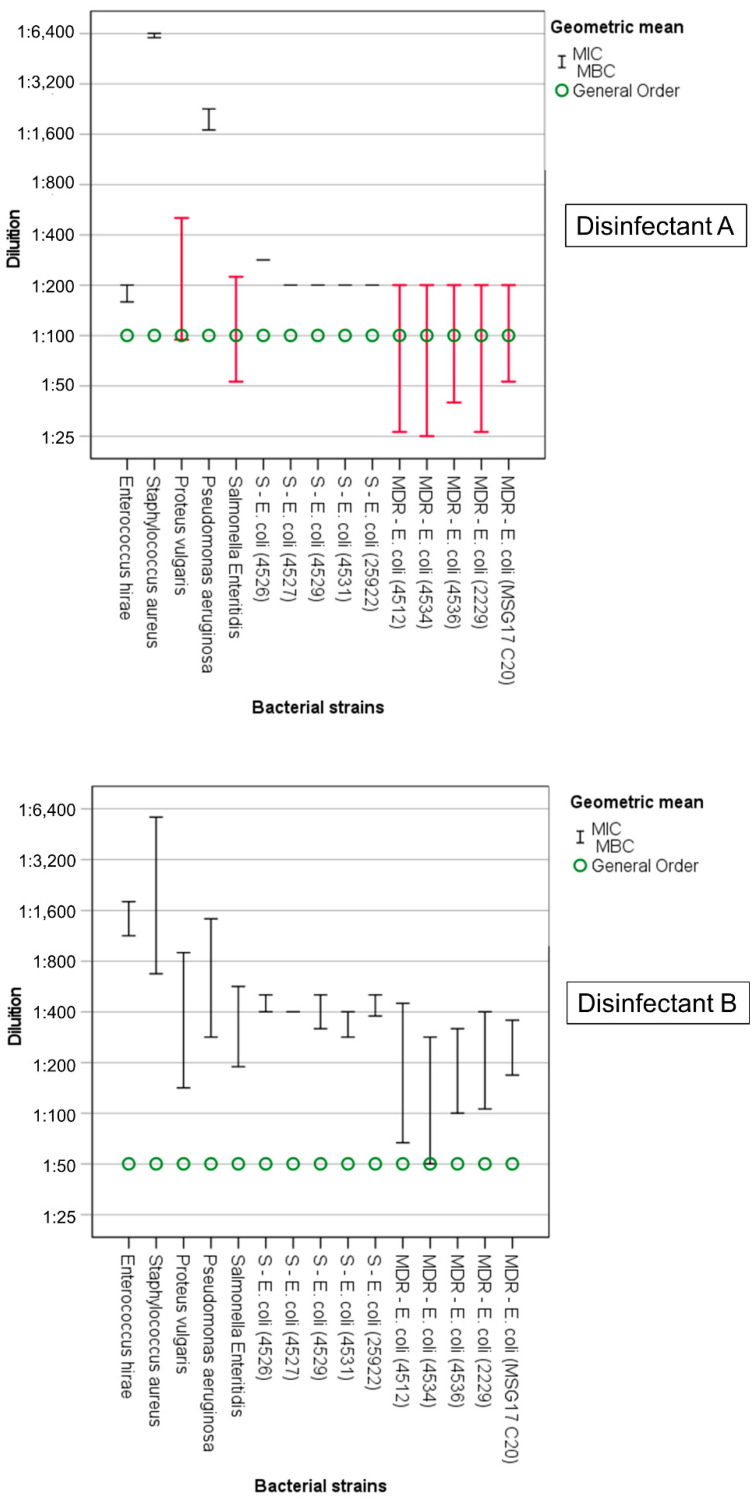
Disinfectant A and disinfectant B. Minimum inhibitory concentration (MIC, upper whisker) and minimum bactericidal concentration (MBC, lower whisker). In red, bacterial strains tolerant to disinfectants (MBC > GO).

**Figure 2 animals-12-02780-f002:**
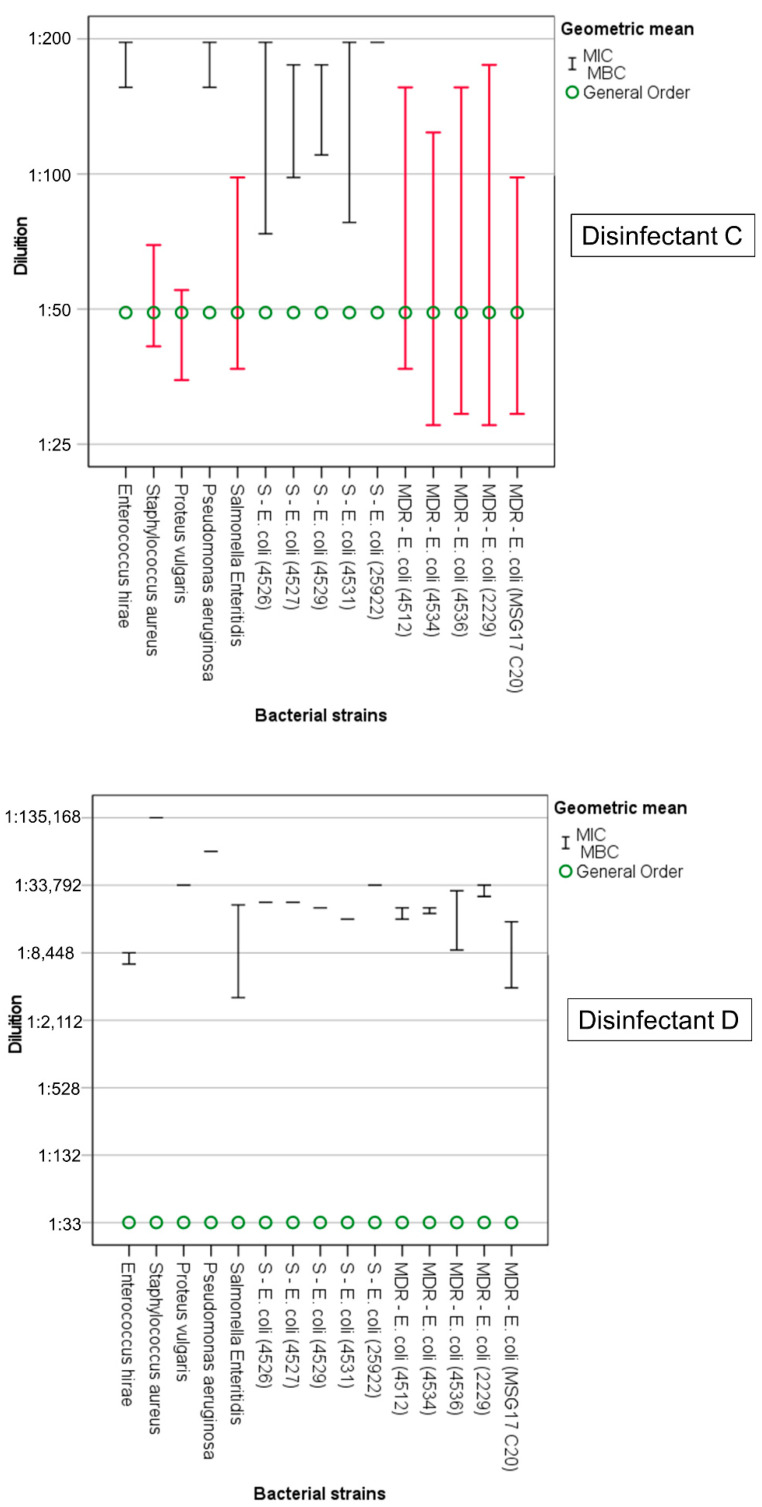
Disinfectant C and disinfectant D. Minimum inhibitory concentration (MIC, upper whisker) and minimum bactericidal concentration (MBC, lower whisker). In red, bacterial strains tolerant to disinfectants (MBC > GO).

**Figure 3 animals-12-02780-f003:**
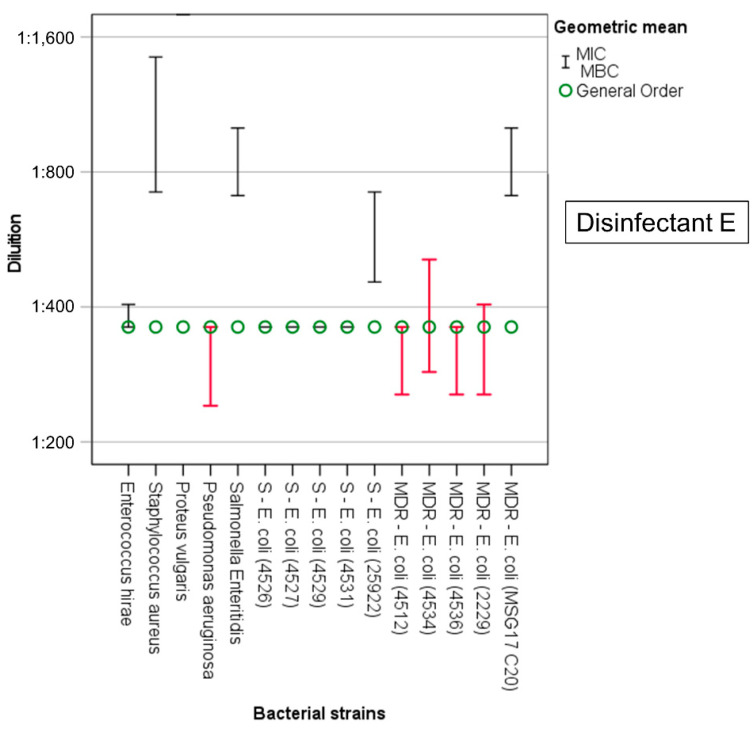
Disinfectant E. Minimum inhibitory concentration (MIC, upper whisker) and minimum bactericidal concentration (MBC, lower whisker). In red, bacterial strains tolerant to disinfectants (MBC > GO).

**Table 1 animals-12-02780-t001:** Antimicrobial resistances of the *Escherichia coli* strains selected for the study.

*E. coli* Strain	AMP	AZM	CTX	CAZ	CHL	CIP	CST	GEN	MEM	NAL	SFX	TET	TIG	TMP	MDR	Phenotype
2229	R	S	R	R	R	R	R	R	S	R	R	R	S	R	Yes	
4512	R	S	R	R	R	R	S	S	S	S	R	R	S	R	Yes	ESBL
4526	S	S	S	S	S	S	S	S	S	S	S	S	S	S	No	
4527	S	S	S	S	S	S	S	S	S	S	S	S	S	S	No	
4529	R	S	S	S	S	S	S	S	S	S	S	R	S	S	No	
4531	S	S	S	S	S	S	S	S	S	S	S	S	S	S	No	
4534	R	S	R	R	R	R	S	R	S	R	R	R	S	R	Yes	AMPC
4536	R	S	R	R	S	S	S	S	S	S	R	S*	S	R	Yes	ESBL/AMPC
ATCC 25922	S	S	S	S	S	S	S	S	S	S	S	S	S	S	No	
MSG17 C20	R	nd	R	nd	nd	R	S	S	S	R	R	R	nd	R	Yes	

AMP: ampicillin; AZM: azithromycin; CTX: cefotaxime; CAZ: ceftazidime; CHL: chloramphenicol; CIP: ciprofloxacin; CST: colistin; GEN: gentamicin; MEM: meropenem; NAL: nalidixic acid; SFX: sulfamethoxazole; TET: tetracycline; TIG: tigecycline; TMP: trimethoprim; R: resistant; S: sensitive; S*: intermediate; MDR: multidrug-resistant; ESBL: extended spectrum β-lactamase; AMPC: AmpC β-lactamases; nd: not done.

**Table 2 animals-12-02780-t002:** Characteristics of the disinfectants used and their recommended concentrations (as of March 2020).

Product	Disinfectant Category	Active Ingredient(s)	General Orders Concentration	Initial Concentration ^a^
Disinfectant A	Peroxygen	Potassium peroxymonosulfate, sodium chloride	100 ^b^	25 400 ^d^
Disinfectant B	Phenol based	Chlorocresol, propionic acid, phosphoric acid	50 ^c^	25 400 ^e^ 100 ^f^
Disinfectant C	Halogen-releasing agent	Iodine, sulphuric acid, phosphoric acid	49 ^c^	24.5
Disinfectant D	Glutaralaldehyde and quaternary ammonium compound	Alkyl dimethyl benzyl ammonium chloride, didecyl dimethyl ammonium chloride, glutaraldehyde	33 ^c^	ND ^g^
Disinfectant E	Halogen-releasing agent	Troclosene sodium (NaDCC)	360 ^b^	180 500 ^h^

^a^ Initial concentration: expressed in the table as a dilution rate, it represents the maximum concentration of disinfectant tested, from which a two fold dilution started; ^b^ milliliters (mL) of water/solvent per gram of product; ^c^ mL of water/solvent per mL of product; ^d^ initial concentration against *Staphylococcus aureus* and *Pseudomonas aeruginosa;*
^e^ initial concentration against *Staphylococcus aureus*; ^f^ initial concentration against *Enterococcus hirae*; ^g^ not definable; ^h^ initial concentration against *Salmonella* Enteritidis and *Escherichia coli* strain MSG17 C20.

**Table 3 animals-12-02780-t003:** Tolerance profiles of the bacterial strains against disinfectants.

Bacterial Strain	Disinfectant A	Disinfectant B	Disinfectant C	Disinfectant D	Disinfectant E
*Salmonella* Enteritidis	T	ES	T	ES	S
*Escherichia coli* strain MSG17 C20 (MDR)	T	ES	T	ES	S
*Escherichia coli* strain 4534 (MDR)	T	S	T	ES	T
*Escherichia coli* strain 4536 (MDR)	T	PS	T	ES	T
*Escherichia coli* strain 4512 (MDR)	T	S	T	ES	T
*Escherichia coli* strain 2229 (MDR)	T	PS	T	ES	T
*Proteus vulgaris*	T	PS	T	ES	ES
*Staphylococcus aureus*	S	ES	T	ES	PS
*Enterococcus hirae*	S	ES	ES	ES	S
*Pseudomonas aeruginosa*	S	ES	ES	ES	T
*Escherichia coli* strain 25922	S	ES	ES	ES	S
*Escherichia coli* strain 4526	S	ES	S	ES	S
*Escherichia coli* strain 4527	S	ES	S	ES	S
*Escherichia coli* strain 4529	S	ES	PS	ES	S
*Escherichia coli* strain 4531	S	ES	S	ES	S

T (tolerant): MBC > GO (recommended in-use concentration); S (sensitive): MBC ≤ GO; PS (particularly sensitive): MBC ≤ 2 × GO; ES (extremely sensitive): MBC ≤ 3 × GO; MDR: multidrug-resistant.

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
