# Peer review of "Efficacy of Five Disinfectant Products Commonly Used in Pig Herds against a Panel of Bacteria Sensitive and Resistant to Selected Antimicrobials"

_animals, 2022, doi:10.3390/ani12202780_

Round 1
Reviewer 1 Report
The manuscript entitled “Efficacy of five disinfectant products commonly used in pig herds against a panel of bacteria sensitive and resistant to selected antimicrobials” addresses a topic of considerable interest and topicality, especially in view of the growing phenomenon of antibiotic resistance. It is a well-structured and well-written manuscript with attention to detail in each of its sections.
I commend the authors for the clarity of presentation of the 'Materials and Methods' section and the results, especially the graphic display of the results, which is immediate and makes it very easy to make comparisons between the effects of the various disinfectants.
The discussion of the results obtained is articulated and supported by an adequate and recent bibliography.
The addition of the supplementary material speaks in favour of the completeness of the investigation carried out and the extreme scientific rigour demonstrated by the authors, whose commitment I hope will be rewarded by the speedy publication of the article they produced.
Author Response
Dear Reviewer, on behalf of all the Authors, I would like to express my sincere gratitude to you for your positive comments.
Reviewer 2 Report
It was a pleasure to read the manuscript testing different disinfectants against field strains of bacteria. It is rare to see a study like this, with so many studies focusing on drug resistance but very few looking at disinfectants which are used much more commonly on farming environment. So it was nice to see a study like this conducted, and conducted well too.
The manuscript is well executed, and well written. Like every reviewer, we would always love to see more bacteria tested but its not always feasible. I only have some very minor comments, and these are entirely up to the authors if they wish to take them on board or not.
Line 63- maybe reduction in antimicrobial usage would sound better?
Having undertaken similar studies myself, I am always concerned that the disinfectant used may have an adverse effect on the media. Did you see any evidence of that here? I am guessing that you ran positive controls which worked throughout it all?
Your results section is a bit ‘wordy’ I wonder if a table (maybe a large one) with the MIC and MBC for each bacteria and highlight in bold those which are over the GO value? It may make it easier for a reader. Again, this is just a thought, and not a requirement. I will leave that to the authors.
I also wonder if the figures would be easier to compare if they were in one block, as A-E? Again, just a suggestion
But overall, a very nice and interesting manuscript which I thoroughly enjoyed reading. My thanks, and best wishes for the future
Author Response
It was a pleasure to read the manuscript testing different disinfectants against field strains of bacteria. It is rare to see a study like this, with so many studies focusing on drug resistance but very few looking at disinfectants which are used much more commonly on farming environment. So it was nice to see a study like this conducted, and conducted well too.
The manuscript is well executed, and well written. Like every reviewer, we would always love to see more bacteria tested but its not always feasible.
Answer: The Authors thank the Reviewer for the positive comments on the paper and for the suggestions provided.
I only have some very minor comments, and these are entirely up to the authors if they wish to take them on board or not.
Line 63- maybe reduction in antimicrobial usage would sound better?
Answer: The Authors agree with the Reviewer’s comments, and the sentence at line 63 has been changed.
Having undertaken similar studies myself, I am always concerned that the disinfectant used may have an adverse effect on the media. Did you see any evidence of that here? I am guessing that you ran positive controls which worked throughout it all?
Answer: Positive controls were used throughout the experiments and levels of growth between test and controls were evaluated to highlight any issue.
Your results section is a bit ‘wordy’ I wonder if a table (maybe a large one) with the MIC and MBC for each bacteria and highlight in bold those which are over the GO value? It may make it easier for a reader. Again, this is just a thought, and not a requirement. I will leave that to the authors.
Answer: A further supplementary table (table S2) has been produced to help addressing the Reviewer’s comment.
I also wonder if the figures would be easier to compare if they were in one block, as A-E? Again, just a suggestion
Answer: The Authors considered the Reviewer’s comment but feel that figures in one block would be smaller and therefore more difficult to read, and prefer to leave the figures in the same format as the original submission.
But overall, a very nice and interesting manuscript which I thoroughly enjoyed reading. My thanks, and best wishes for the future